# Pectin–Chitosan Hydrogel Beads for Delivery of Functional Food Ingredients

**DOI:** 10.3390/foods13182885

**Published:** 2024-09-12

**Authors:** Eduardo Morales, Marcela Quilaqueo, Rocío Morales-Medina, Stephan Drusch, Rodrigo Navia, Agnès Montillet, Mónica Rubilar, Denis Poncelet, Felipe Galvez-Jiron, Francisca Acevedo

**Affiliations:** 1Scientific and Technological Bioresource Nucleus (BIOREN), Universidad de La Frontera, Avenida Francisco Salazar, Temuco 01145, Chile; eduardo.morales@ufrontera.cl (E.M.); marcela.quilaqueo@ufrontera.cl (M.Q.); monica.rubilar@ufrontera.cl (M.R.); 2Department of Chemical Engineering, Faculty of Engineering and Sciences, Universidad de La Frontera, Casilla 54-D, Temuco 4811230, Chile; rodrigo.navia@ufrontera.cl; 3Department of Food Technology and Food Material Science, Institute of Food Technology and Food Chemistry, Technische Universität Berlin, Königin-Luise-Str. 22, 14195 Berlin, Germany; morales-medina@tu-berlin.de (R.M.-M.);; 4Centre for Biotechnology and Bioengineering (CeBiB), Faculty of Engineering and Sciences, Universidad de La Frontera, Casilla 54-D, Temuco 4811230, Chile; 5Oniris, CNRS, GEPEA, Nantes Université, UMR 6144, F-44600 Saint-Nazaire, France; agnes.montillet@univ-nantes.fr; 6EncapProcess, 114 Allée Paul Signac, F-44240 Sucé sur Erdre, France; denis.poncelet@encapprocess.fr; 7Doctoral Program in Sciences with a Specialty in Applied Cellular and Molecular Biology, Faculty of Medicine, Universidad de La Frontera, Temuco 4811230, Chile; felipe.galvez.j@gmail.com; 8Department of Basic Sciences, Faculty of Medicine, Universidad de La Frontera, Casilla 54-D, Temuco 4811230, Chile; 9Center of Excellence in Translational Medicine (CEMT), Faculty of Medicine, and Scientific and Technological Bioresource Nucleus (BIOREN), Universidad de La Frontera, Casilla 54-D, Temuco 4811230, Chile

**Keywords:** ionic gelation, interfacial coacervation, encapsulation, hydrogel, coating, controlled release

## Abstract

A common challenge in hydrogel-based delivery systems is the premature release of low molecular weight encapsulates through diffusion or swelling and reduced cell viability caused by the low pH in gastric conditions. A second biopolymer, such as chitosan, can be incorporated to overcome this. Chitosan is usually associated with colonic drug delivery systems. We intended to formulate chitosan-coated pectin beads for use in delaying premature release of the encapsulate under gastric conditions but allowing release through disintegration under intestinal conditions. The latter is of utmost importance in delivering most functional food ingredients. Therefore, this study investigated the impact of formulation and process conditions on the size, sphericity, and dissolution behavior of chitosan-coated hydrogel beads prepared by interfacial coacervation. The size and sphericity of the beads depend on the formulation and range from approximately 3 to 5 mm and 0.82 to 0.95, respectively. Process conditions during electro-dripping may be modulated to tailor bead size. Depending on the voltage, bead size ranged from 1.5 to 4 mm. Confocal laser scanning microscopy and scanning electron microscopy confirmed chitosan shell formation around the pectin bead. Chitosan-coated beads maintained their size and shape in simulated gastric fluid but experienced structural damage in simulated intestinal fluid. Therefore, they represent a novel delivery system for functional food ingredients.

## 1. Introduction

Hydrogel beads for delivery of functional food ingredients are mainly based on ionotropic gelation of alginate. Low methyl-esterified pectin may also easily be crosslinked by calcium ions to form hydrogels [1,2]. Pectin microcapsules can be produced using a process similar to that of alginate beads. Similar to alginate, pectin is broken down by enzymes produced by the microbes in the colon. The benefits of pectin are widely recognized. Some years ago, Liu et al. [3] stated that “the commercial potential of this technology (i.e., pectin-based hydrogel beads) has yet to be realized”. The authors explain that the lack of reproducible performance and the large diversity in pectin’s molecular characteristics were the major challenges at that time. Nowadays, a range of well-standardized pectin qualities with tailored functionality is widely available at a competitive cost for food applications. Pectin is thus considered an ideal delivery system for probiotic cell encapsulation due to its strong film-forming and binding abilities [3] and its stability and favorable microenvironment for cell growth [4]. Furthermore, there has been intense research on immunomodulating properties and the impact of pectin on gut barrier function, showing its positive and synergistic effect on physiological functionality [5,6,7].

However, hydrogel beads are porous and contain a significant amount of water. This is an unambiguous prerequisite in some applications, like the immobilization of yeasts, bacteria, or enzymes for biotechnological applications. Conversely, when considering the delivery of nutrients or drugs to the lower sections of the gastrointestinal tract, it is important to prevent the unintentional release of the encapsulated substance from the hydrogel due to diffusion or swelling of the hydrogel beads. Additionally, it is crucial to prevent a decrease in the viability of probiotic cells in the presence of gastric conditions caused by the low pH. Two main strategies exist to overcome this limitation. The first option is incorporating a second biopolymer to produce a composite matrix with the desired functionality; another option is using an interfacial coacervation with a second biopolymer to achieve coating for modification of functionality.

In this context, chitosan is a promising candidate, raising particular interest in pectin/chitosan-based drug delivery systems [1]. The positively charged amino group of chitosan can interact through electrostatic interactions with the negatively charged groups of pectin, namely those of the galacturonic acid backbone, to form complex coacervates or composite particles through intermolecular interactions or even a joined polymeric network. Pectin–chitosan-based composite materials were used to protect quercetin against premature release under gastric conditions [8] or delayed release of the antimicrobial peptide nisin for the stabilization of foods [9]. Dziadek et al. [1] showed a higher mechanical strength for a dried pectin–chitosan composite material compared to the pectin-based material. A higher tensile strength has also been reported for pectin–chitosan-based membranes, especially when crosslinked with calcium [10]. Chitosan-composite materials are not an option for cell encapsulation, mainly due to electrostatic interactions with negatively charged cell membrane groups. This mechanism causes intracellular material to leak, which is the basis of the antibacterial activity of chitosan [11]. Rashidova et al. [12] observed an antifungal activity with no growth of *A. niger* in the pectin/chitosan matrix, while there was significant growth in a pure pectin matrix.

Thus, one may prefer to use chitosan as a coating material for the polar surface of carbohydrate-based hydrogel beads, where electrostatic interactions significantly contribute to adhesion. Delivery systems resulting from interfacial coacervation show a core–shell-type structure with a hydrogel bead as a core. Belscak-Cvitanovic et al. [13] describe a significantly reduced diffusion of caffeine from alginate beads coated with chitosan in an aqueous environment. A system in which chitosan particles were coated with pectin was used to retard curcumin release compared to composite particles consisting of the same materials [14]. More recently, Sampathi [15] prepared chitosan-coated beads containing a pectin-containing curcumin nanosuspension using ionotropic gelation with zinc chloride. To the best of our knowledge, only a very limited number of studies are available that investigate the impact of formulation and processing on bead characteristics and functionality for chitosan-coated hydrogel beads. Das [16] examined the effect of pH and chitosan content during crosslinking, crosslinking time, and polymer–drug ratio on the release of resveratrol from chitosan-coated pectin beads gelled using zinc acetate. Increased chitosan content and crosslinking time reduced the premature release of encapsulates under gastric conditions. Furthermore, there is evidence from other hydrogel systems that particle size and shape uniformity affect release behavior [17].

To sum up, it is well described in the literature that the combination of pectin and chitosan protects against premature release of encapsulates under gastric conditions. However, there is less evidence on the impact of formulation and processing on the characteristics of the hydrogel beads and whether the presence of chitosan affects hydrogel bead disintegration and release of the encapsulate. Therefore, the present study aimed to investigate the effect of chitosan and pectin type, their content, and calcium concentration during hardening on hydrogel bead size, sphericity, and swelling behavior. A better understanding will allow us to tailor hydrogel bead properties to its functionality since particle characteristics like size, sphericity, and surface structure affect release properties.

## 2. Materials and Methods

Two samples of low methyl-esterified citrus pectin were kindly provided by CP Kelco A/S (Lille Skensved, Denmark). According to the manufacturer’s specification, one sample had a degree of methyl-esterification of 30% (P30), and the other sample had a degree of methyl-esterification of 35% (P35). More importantly, the samples also differed in calcium sensitivity, indicating a difference in the distribution of the free carboxylic groups in the galacturonic acid backbone. P30 showed a calcium sensitivity of 253.8 mPa·s, while P35 showed a calcium sensitivity of 476.5 mPa·s, as determined by Archut et al. [18]. Briefly, calcium sensitivity is determined by subtracting the viscosity of the pectin sample in the presence and absence of a defined amount of calcium ions.

Two fungal chitosan samples ((C_6_H_11_NO_4_)n) were obtained from KitoZyme S.A. (Herstal, Belgium). According to the manufacturer’s specification, KiOsmetine^®^ CsH (CsH) has a degree of acetylation below 20%, a high molecular weight (average: 60 kDa), and is derived from the fungus *Agaricus bisporus*. KiOsmetine^®^ CsG (CsG), derived from the mycelium of *Aspergillus niger*, is characterized by a low molecular weight (average: 20 kDa) and also a degree of acetylation of less than 20%. Calcium chloride dihydrate (CaCl_2_·2H_2_O) and glacial acetic acid were purchased from VWR International LLC.

### 2.1. Preparation of Solutions

Calcium chloride solution was prepared by dissolving CaCl_2_·2H_2_O in deionized water (concentrations between 0 and 130 mM) under magnetic stirring. Chitosan solutions at different concentrations (0.5–3 wt.%) were dissolved in calcium chloride solution (0–130 mM) and 0.5 wt.% acetic acid under gentle stirring at ambient temperature until complete dissolution. Pectin solutions were prepared at different concentrations (0.5–3 wt.%) by dissolving them in deionized water under magnetic stirring at 50 °C for 3 h. All solutions were stored at 4 °C until used at room temperature.

### 2.2. Viscosity Measurements

The shear viscosity of the pectin solutions and chitosan/calcium chloride solutions was determined using a rheometer (Physica MCR 500 or MCR 102e, Anton Paar GmbH, Ostfildern, Germany) equipped with a concentric cylinder geometry or a parallel plate 50 mm in diameter, depending on samples. The sample placed in the cylinder was subjected to an increase in shear rate from 1 to 500 s^−1^ with 30 measurement points and 10 s at each point. During the viscosity measurements, the temperature was kept at 20 °C.

### 2.3. Zeta Potential Measurements

Zeta potential measurements were performed on a zetasizer (Zetasizer Nano ZS, Malvern Panalytical Ltd., Sao Paolo, Brazil) using disposable folded capillary cells DTS1070. Chitosan (CsH) and pectin solutions (P30, P35) were prepared at 0.05 wt.% with distilled water. The pH was adjusted using 0.1 M and 1 M hydrochloric acid or sodium hydroxide solutions. Analysis was performed with three independent replicates and trifold instrumental analysis. Temperature was fixed at 25 °C, the refractive index of the dispersant was 1.333, and the viscosity was set at 0.889 mPa·s. The refractive index of the material was set to 1.450, and dispersant viscosity was used as sample viscosity.

### 2.4. Screening for Selection of Polymers, Polymer Content, and Calcium Concentration for Hydrogel Bead Production

Preliminary assays were performed to evaluate the formation of beads using both types of pectin at 1.5 wt.% and chitosan in a range of 0.5–3.0 wt.%. Chitosan solutions containing 100 mM of CaCl_2_·2H_2_O were used. The pectin solution was extruded using a syringe with an internal needle diameter of 2 mm above a beaker containing the chitosan solution. The distance between the surface of the solution and the syringe was 5 cm. The pectin solution was dropped with a syringe (10 mL capacity) into 50 mL of chitosan and calcium chloride solution. Gel beads were formed upon contact with the bath solution and were left to harden for 30 min. After hardening, the beads were removed from the solution by simple filtration using a sieve of appropriate mesh size. The optical evaluation was performed based on the size and sphericity of the beads.

### 2.5. Characterization of the Effect of Polymer Content and Calcium Concentration on Hydrogel Bead Size and Sphericity

A Box–Behnken experimental design was used to further characterize the effect of the abovementioned factors on hydrogel bead size and sphericity (Table 1). pH was not adjusted, and bead formation was carried out at room temperature. Experiments were carried out in a randomized order to minimize the effect of unexpected variability on the observed response due to extraneous factors.

A second-order model (Equation (1)) was used to describe response variable mean particle size and sphericity factor (SF) as a function of coded content of pectin (X_1_: Pectin), content of chitosan (X_2_: Chitosan), and concentration of CaCl_2_·2H_2_O (X_3_: CaCl_2_):SF = A_0_ + A_1_X_1_ + A_2_X_2_ + A_3_X_3_ + A_11_X_11_ + A_22_X_22_ + A_33_X_33_ + A_12_X_1_X_2_ + A_13_X_1_X_3_ + A_23_X_2_X_3_(1)
where A_0_ is a constant; A_1_, A_2_ and A_3_ are linear coefficients; A_11_, A_22_, and A_33_ are quadratic coefficients; A_12_, A_13_ and A_23_ are cross-product coefficients. The experimental data were subject to multiple regression analysis to obtain the adjusted polynomial equation. An ANOVA was used to identify the significant coefficients. Finally, contour plots were used to visualize the effect of the independent variables. Design Expert was used to analyze the results. All experiments were performed in triplicate.

The mean size of beads and sphericity were evaluated from images acquired by a digital camera and analyzed by ImageJ 1.45s software ( National Institutes of Health, Bethesda, MD, USA). Approximately 25 beads were measured to determine the following parameters:Sphericity factor (SF) = 1 − (D_max_ − D_min_)/(D_max_ + D_min_)(2)
Mean size (M_size_) = (D_max_ + D_min_)/2(3)
where D_max_ is the maximum diameter of the beads, and D_min_ is the perpendicular diameter of D_max_.

### 2.6. Microbead Formation Using Electrostatic Droplet Generation (Electro-Dripping)

Based on the Box–Behnken experimental design results, microbeads were produced using a self-made electro-dripping system varying the voltage from 0 to 10 kV. The pectin solution (2.2 wt.%) was extruded from a syringe (35 mL capacity) through a needle connected to an electrostatic energy generator using a syringe pump (World Precision Instruments, Sarasota, FL, USA) at a flow rate set at 3 mL min^−1^. This generator was placed above a beaker containing 50 mL of chitosan solution (2.2 wt.% in 130 mM CaCl_2_·2H_2_O) and connected to the negative but grounded connector of the electrostatic generator. The needle (internal size 616 µm, external size 950 µm) was fixed on a self-made 3D-printed build block through luer lock connectors. The distance between the surface of the solution and the needle was 20 cm.

Gel beads formed upon contact with the bath solution and were left to harden for 30 min. After hardening, the beads were removed from the solution by filtration. The sphericity factor and mean size of beads were determined as described above. Images were taken using a Dino-Lite Digital Microscope (Dino-Lite Europe, Almere, The Netherlands) and confirmed under a VHX-7000 digital microscope (Keyence France, SAS, Bois Colombes, France).

### 2.7. Characterization of Hydrogel Bead Particle Morphology and Structure

Microbeads were characterized by scanning electron microscopy. The beads were placed on 10 mm copper tape (3 M) and mounted on a brass disk to visualize particle morphology in the wet state. The samples were analyzed using magnifications of 250× and 500× under low vacuum conditions using either a SU3500 Hitachi scanning electron microscope (Hitachinaka, Japan) at an acceleration voltage of 10 kV or an MEB ZEISS EVO 40, operated in partial pressure mode of 34 Pa and accelerating voltage 15 kV.

In addition, the chitosan layer was visualized using confocal laser scanning microscopy. For this purpose, chitosan (250 mg) was dissolved in 25 mL of 2 vol.% acetic acid, and 10 mg of fluorescein isothiocyanate (FITC) was dissolved in 1 mL of ethanol. Both solutions were mixed and shaken for 24 h in the dark at 37 °C to ensure molecular binding of the marker to chitosan. The mixture was then purified by dialysis (Membra-Cel MC18*100CLR, molecular weight cut-off of 14 kDa) for 72 h, changing the medium every 24 h to ensure the removal of FITC not bound to chitosan. Microbeads were prepared as described above using the FITC-labeled chitosan. The beads were analyzed by confocal microscopy, using an FV1000 Olympus, to study the distribution of chitosan in the bead. The excitation wavelength was 488 nm, and the emission wavelength was 533 nm.

### 2.8. Characterization of the Swelling Behavior of Hydrogel Beads under Simulated Gastrointestinal Conditions

Pectin-based hydrogel beads with and without chitosan coating were prepared to evaluate the swelling behavior. Pectin P35 was used at a content of 2.2 wt.%. The solution was dripped into a calcium chloride solution (130 mM) or a 2.2 wt.% chitosan CsH solution containing 130 mM calcium chloride. Hydrogel bead preparation was performed as described in Section 2.4. For both samples, the swelling was studied in simulated gastric fluid and subsequently in simulated intestinal fluid for up to 3 h in each medium with sampling every 30 min. The media matched the pH and electrolyte composition as described in the INFOGEST protocol, but no enzymes were included [19]. Hydrogel bead size and sphericity were determined as described in Section 2.5.

## 3. Results and Discussion

In the first step, the shear viscosity and zeta potential of two pectin (P30 and P35) and two chitosan solutions (CsG, CsH) were investigated. Also, their potential to form hydrogel beads (by simple dripping) was screened with a visual evaluation of bead characteristics. Based on these experiments, types of pectin and chitosan were chosen to investigate the impact of their content and calcium concentration in the crosslinking solution on hydrogel bead size and sphericity. In a third step, to ensure transfer into pilot and production scale, manual dripping was changed to electrostatic droplet generation to characterize the impact of the applied voltage on bead size and sphericity. Finally, the pH-dependent swelling behavior of chitosan-coated pectin beads was analyzed to gain insight into the functionality change under simulated gastrointestinal conditions.

### 3.1. Viscosity and Zeta Potential of Pectin and Chitosan Solutions

Pectin samples readily dissolved in water and showed a typical shear-thinning behavior. Viscosity at 100 s^−1^ amounted from 9.5 and 9.8 to 1085 and 1201 mPa·s for a pectin content ranging from 0.5 to 3.0% for P30 and P35, respectively (Figure 1). 

The solution of chitosan CsG was not homogeneous and tended to separate into two phases. Consequently, the system determined a rather low viscosity of 2.3 to 21.1 mPa·s. Viscosity for CsH also showed shear-thinning and ranged from 4.5 to 302.0 mPa·s for 0.5 to 3.0 wt.% at a shear rate of 100 s^−1^ (Figure 2). Since CsG still has a high degree of acetylation, it is not surprising that solubility in dilute acidic solution is low. In CsG, the number of free primary amino groups is rather low; however, protonation of a significant number of these groups would be required for intermolecular electrostatic repulsion, molecular unfolding, and solvation. It is generally known that the solubility of chitosan depends on the degree of acetylation (DA), distribution of acetyl groups, and degree of polymerization [20]. Solubility in a dilute acidic solution requires a degree of deacetylation (DD) of 28%, and solubility in water requires a DD of roughly 50% [21]. In the present study, chitosan CsH with a DA below 30% readily dissolved.

Zeta potential showed pH dependency, as expected from the chemical structure of pectin and chitosan. The zeta potential of both pectin solutions decreased from approximately −15 mV to −50 mV and from pH 2 to pH 4, respectively (Figure 3). The pK_a_ of the free carboxylic groups of galacturonic acid is 3.5 [22]. In contrast, the zeta potential of chitosan decreased from 55 to 36 mV upon increasing the pH in the same range. The pK_a_ value of the primary amino groups in chitosan is 6.3. Consequently, it is reasonable to expect a substantial positive net charge within the pH range examined in the current study.

### 3.2. Screening for Selection of Polymers, Polymer Content, and Calcium Concentration for Hydrogel Bead Production

Screening experiments showed that only two of the materials (P35 and CsH) interacted in a way that hydrogel beads were formed under the chosen conditions. In the case of pectin, no bead formation was observed for P30, regardless of the type of chitosan, pectin, or calcium concentration in the crosslinking solution, which was up to 1.5 wt.%.

Dripping of pectin P35 into the crosslinking solution led to the formation of beads. However, marked differences between the two different chitosan samples were observed. When using Chitosan CsG with low molecular weight, no beads were formed at 0.5 wt.% and 3.0 wt.% chitosan in the crosslinking solution. In the range of 1.0 to 2.0 wt.%, nonspherical beads were formed. In contrast, when using chitosan CsH with a high molecular weight, spherical hydrogel beadlets were obtained with a chitosan content between 1.0 and 3.0 wt.%. The crosslinking solution had the highest sphericity between 1.0 and 2.2 wt.% chitosan. As reported by Prüsse et al. [23], the viscosity of the crosslinking solution can significantly influence the shape of the polymer droplets. High viscosities (>150 mP·s at a shear rate above 100/s) obtained with a CsH concentration higher than 2.2 wt.% can cause beads to deform. In addition, lower viscosities do not lead to bead formation, as reported by Santos et al. and Moghadam et al. [24,25]. Based on this screening and reasoning, pectin P35 and chitosan CsH were selected for the next experiments.

### 3.3. Characterization of the Effect of Polymer Content and Calcium Concentration on Hydrogel Bead Size and Sphericity in Simple Dripping

In a Box–Behnken design, the effect of pectin, chitosan content, and calcium chloride concentration on hydrogel bead size and sphericity was analyzed. Both parameters are important characteristics of hydrogel beads. In many applications, hydrogel beads should be small to allow rapid exchange with the external medium and diffusion of nutrients. This is particularly important in processes such as cell encapsulation or enzyme mobilization. At the same time, irregular shapes, protrusions, or tails need to be avoided since they may cause immune responses in biomedical applications [25].

The effect of pectin content and calcium concentration on hydrogel bead size is depicted in Figure 4A–C. Concerning hydrogel bead size, a linear model was most suitable to describe the data (Table 2). With a *p*-value of 0.0020 and a lack of fit of 0.3402, the model is significant. The adjusted R^2^ amounted to 0.6513, and the predicted R^2^ to 0.4571. When choosing a higher-order model, the sequential *p*-value became insignificant, and the predicted R^2^ decreased. Mean hydrogel bead size ranged from 3.0 ± 0.1 to 4.5 ± 0.1 mm and was significantly affected by pectin concentration. Pectin content governs the viscosity of the solution, and the latter is crucial in many ways during processing [25,26,27]. However, drop generation at the tip of the needle in simple dripping is mainly determined through gravitational force and surface tension of the solution. At this stage, viscosity is not relevant in the range of 10 to 100 mPa·s [27], and more recently, the same group reported similar droplet sizes for alginate solutions even with a viscosity of up to 400 mPa·s [26]. Furthermore, the viscosity of the pectin solutions did not differ that much in the present study. Considering that hydrogel bead size results from drop size and subsequent change in size during crosslinking, the latter must, therefore, be the major reason for the difference in hydrogel bead size.

The effect of pectin content and calcium concentration on hydrogel bead sphericity is depicted in Figure 4D–F. Data on sphericity were described best using a quadratic model. The sequential *p*-value was 0.0282, and the next hierarchical level, the cubic level, is aliased. The lack of fit was insignificant (0.3175), and predicted and adjusted R^2^ amounted to 0.5599 and 0.9045, respectively (Table 3). It becomes obvious that also concerning sphericity, pectin content of the solution, and thus viscosity is the major factor in the present study. First of all, droplets in simple dripping initially have a tear-like shape. Surface tension is the driving force for the spherical shape, but viscosity counteracts the change in drop morphology [28]. Based on the work of Davarci et al. [26], it must be assumed that a spherical shape is achieved within milliseconds in the viscosity range studied in our work; therefore, irregular drop shape after dripping must be ruled out as an explanation for differences in sphericity. The more important point is deformation occurring upon contact of the droplet with the calcium chloride solution. Kinetic energy must be high enough to break the surface resistance of the solution, and viscosity must be high enough to prevent deformation [26].

### 3.4. Microbead Formation Using Electrostatic Droplet Generation (Electro-Dripping)

Ionic gelation facilitated by electro-dripping can reduce particle size if desired. Another advantage of electrostatic dripping is the uniformity of size distribution and sphericity [23]. By applying an electrical potential, the migration of charged molecules to the surface of the droplet is promoted. Since molecules of similar charge repel each other, surface occupancy and, thus, surface tension decreases. In fact, under these conditions, the electrostatic pressure applied on the surfaces forces the liquid to drop into a cone shape [29]. The surplus charge is ejected by the emission of droplets from the tip of the solution.

In the present study, the most appropriate formulation from the previous experiment (2.2 wt.% pectin, 2.2 wt.% chitosan, and 130 mM calcium chloride) was used for the microbead formation to generate electrostatic droplets at different voltages between 0 and 9.8 kV. The results showed a reduction in the droplet size when the electrostatic potential increased (Figure 5). The mean bead size decreased from 3.5 mm to 1.5 mm. The latter aligned with our expectation since, at this electrical voltage, the drop size will be approximately double the internal diameter of the needle [25]. Poncelet et al. [27] explained that there is a critical value for the voltage defined as the voltage when unstable liquid jets of polymer solutions begin to form and disaggregate in multiple microscopic charged droplets, whose size does not vary with later increase. In our case, this critical voltage seems to be in the region between 8 and 10 kV. Although the droplet size still decreased, the sphericity value also tends to decrease above 8 kV.

### 3.5. Particle Morphology

Samples from both experiments investigating formulation and electrostatic dripping were characterized by scanning electron microscopy (SEM) and/or confocal laser microscopy (CLSM). As shown in Figure 6A, the beads showed a smooth spherical shape with no pores, but some superficial wrinkles can be noticed on their surface. Confocal microscopic analysis showed the fluorescence of chitosan around the bead (Figure 6B), confirming the presence of this polysaccharide as a coating on the pectin particle.

When using SEM (Figure 6C,D) to characterize the beads from electrostatic dripping at higher magnifications, the surface revealed a rougher surface (Figure 6C), but this is due to a dehydration process linked to the applied vacuum. This process induces a slight shrink of the bead. The striations on the bead’s surface are made more visible, as well as the kind of scar (on the right side of the bead in Figure 6C). An external layer, including the chitosan coating, was clearly visible when the microbead was cut, and its thickness was in the range of 100 µm in the dried form (Figure 6D).

### 3.6. Swelling Behavior under Simulated Gastrointestinal Conditions

Swelling behavior was tested for pectin-based hydrogel beads with and without chitosan coating. Simulated fluids were prepared according to the INFOGEST protocol, but the addition of enzymes was omitted. It was expected that swelling is mainly determined by pH and ion composition. When studying digestion, pancreatin may accelerate encapsulate release through de-esterification of the pectin [30].

Figure 7 shows that particle size was in a similar range of 4 to 5 mm, and the sphericity factor ranged from 0.93 to 0.95. Furthermore, mean particle size slightly changed in both samples upon incubation in simulated gastric fluid (pH 3) and remained fairly constant for 180 min. Based on the literature, shrinkage of pectin-based hydrogel beads under gastric conditions was expected. The typical explanation is that the protonation of dissociated carboxylic groups within the galacturonic acid backbone reduces electrostatic repulsion and, thus, particle shrinkage. This explanation may hold for hydrogel-based beads in pH-adjusted media in cases where stabilization is mainly achieved through hydrogen bonds and hydrophobic interactions, like in the case of high methyl-esterified pectin. In contrast, electrostatic repulsion may be less pronounced in systems stabilized through ionic gelation in the presence of mono- and divalent cations. Some of the dissociated carboxylic groups are involved in gel network formation, and free dissociated carboxylic groups may be shielded by oppositely charged counter ions. Recently, Reichembach et al. reported disaggregation of pectin beads for both high and low methyl-esterified pectin under acidic conditions [31].

Concerning chitosan, lowering the pH will result in a more pronounced protonation of amine groups. In composite hydrogel beads or chitosan-based hydrogels, this might lead to swelling of the hydrogel beads, as it was reviewed by [32]. However, apart from electrostatic interactions, hydrogen bonds and hydrophobic effects additionally stabilize pectin–chitosan complexes [12]. Therefore, there are also studies in which no pH dependency of particle size alterations was observed [33].

However, in the present study, chitosan is used as a coating material and, thus, does not significantly contribute to particle size, as depicted in Figure 6. The integrity of the coating has been described in the literature at a pH of 3.5. A strong association is assumed due to the charge (derived from pKa values) and the observation that rinsing with buffer does not alter a pectin–chitosan film as determined by surface plasmon resonance [34]. Ventura investigated thermal gelation of low methyl-esterified pectin and chitosan at pH 1.5. Adding urea, which is known to break hydrogen bonds, showed that hydrogen bonds play a major role in pectin–chitosan interactions. The addition of NaCl, which shields the remaining charge of pectin, led to changes so that, obviously, electrostatic interactions still occur even at very low pH [35]. Electrostatic interactions between chitosan and pectin in a composite gel matrix were recently confirmed by FT-IR analysis, where a shift of the carboxylate band from 1597 cm^−1^ to 1588 cm^−1^ occurred [31]. Finally, the integrity of the film can also be derived from other studies where the authors have described the reduction in diffusion-based losses of the encapsulate. Wang et al. observed a significantly reduced release of beta-carotene from pectin–chitosan-coated liposomes compared to chitosan-coated liposomes in simulated gastric fluid (SGF) and simulated intestinal fluid (SIF) after three hours [36].

In the present study, hydrogel beads with and without chitosan coating disintegrated during incubation in simulated intestinal fluid (pH 7). While particle size did not change after 60 min, shrinkage and disintegration were observed after 120 min. They fully dissolved when the hydrogel beads were put back into the simulated intestinal medium after size analysis after 120 min. It is worth mentioning that no degradation occurred when incubating the samples in distilled water with a pH adjusted to pH 7. Therefore, the degradation must mainly be attributed to the presence of the ions and the residence time in the simulated intestinal fluids. There is sufficient evidence in the literature to postulate that degradation is caused by monovalent ions that diffuse through the chitosan coating and displace calcium ions in the pectin network. For example, Birch investigated the swelling of dried pectin–chitosan beads in saline phosphate buffer (approx. 150 mM sodium) and observed a swelling ratio of 340% at pH 6 and pH 7.4 with no difference between the two pH [37]. An intestinal burst release of the encapsulate from chitosan-coated alginate hydrogel beads has been reported [38]. More recently, Wong et al. [38] investigated the swelling and release of thymoquinone from Pickering emulsions encapsulated in alginate chitosan beads. The authors report a swelling of 86% and a water uptake of 593% during intestinal digestion in simulated intestinal fluid. The release of thymoquinone was dominated by diffusion via the swelling process based on the Korsmeyer–Peppas model [39]. In a dedicated study investigating the impact of monovalent cations on the release of the encapsulate, Tan et al. observed a significant increase in tocotrienol release from chitosan-coated alginate beads above a concentration of 50 mM NaCl [40].

## 4. Conclusions

In summary, the molecular structure of both pectin and chitosan determines the capacity for hydrogel bead formation. By choosing their content and the calcium concentration in the medium hydrogel bead size, sphericity and characteristics are defined and may be used to optimize functionality. Particle size, which is important for the release profile, may further be tuned using the dripping technology and selecting appropriate parameters related to the technology. Chitosan coating through interfacial coacervation does not significantly affect disintegration characteristics in simulated gastrointestinal fluids.

While chitosan-based hydrogel beads already serve as a colonic delivery system, chitosan-coated pectin beads are well suited for intestinal delivery. In general, the release of the encapsulate from pectin-based hydrogel beads involves diffusion, erosion, and relaxation of the hydrogel matrix. There are already few studies available giving proof of this assumption. Niu et al. investigated the degradation of lactoferrin, which was encapsulated in pectin or pectin coated with chitosan. The authors observed a short period of degradation under gastric conditions since chitosan attracted pepsin, but then degradation was significantly retarded due to spatial and electrostatic complexation [41]. Park et al. observed in an in vivo study that chitosan does not affect lipid digestibility from chitosan-coated emulsion droplets. In the system by Park et al., the chitosan layer was expected to be rather thin [42], as in the present study. However, crosslinking time affects the release of the encapsulate under simulated intestinal conditions. With zinc–pectin–chitosan beads submitted to 30 min of crosslinking, 45% of encapsulated resveratrol was released under simulated intestinal conditions, while less than 10% was released after crosslinking for 120 min. Furthermore, it is worth mentioning that the molecular weight of chitosan affects the swelling and encapsulate release. Ellagic acid release from chitosan-coated hydrogel beads was not affected when using high molecular weight chitosan, while the intestinal release was significantly retarded when using low molecular weight chitosan [43]. Therefore, it is evident that release can be altered throughout the gastrointestinal passage [16]. Since chitosan also shows mucoadhesive properties [44], residence time and, thus, bioavailability of the encapsulate may be further enhanced, providing an additional benefit to this delivery system. Chitosan-coated pectin beads will, therefore, allow a more efficient delivery of micronutrients and probiotics in functional foods and food supplements, as well as gastrointestinal drug delivery. Additional studies are needed to systematically characterize load capacity, release profile, physicomechanical stability during food manufacture, and biological activities like immune responses. 

## Figures and Tables

**Figure 1 foods-13-02885-f001:**
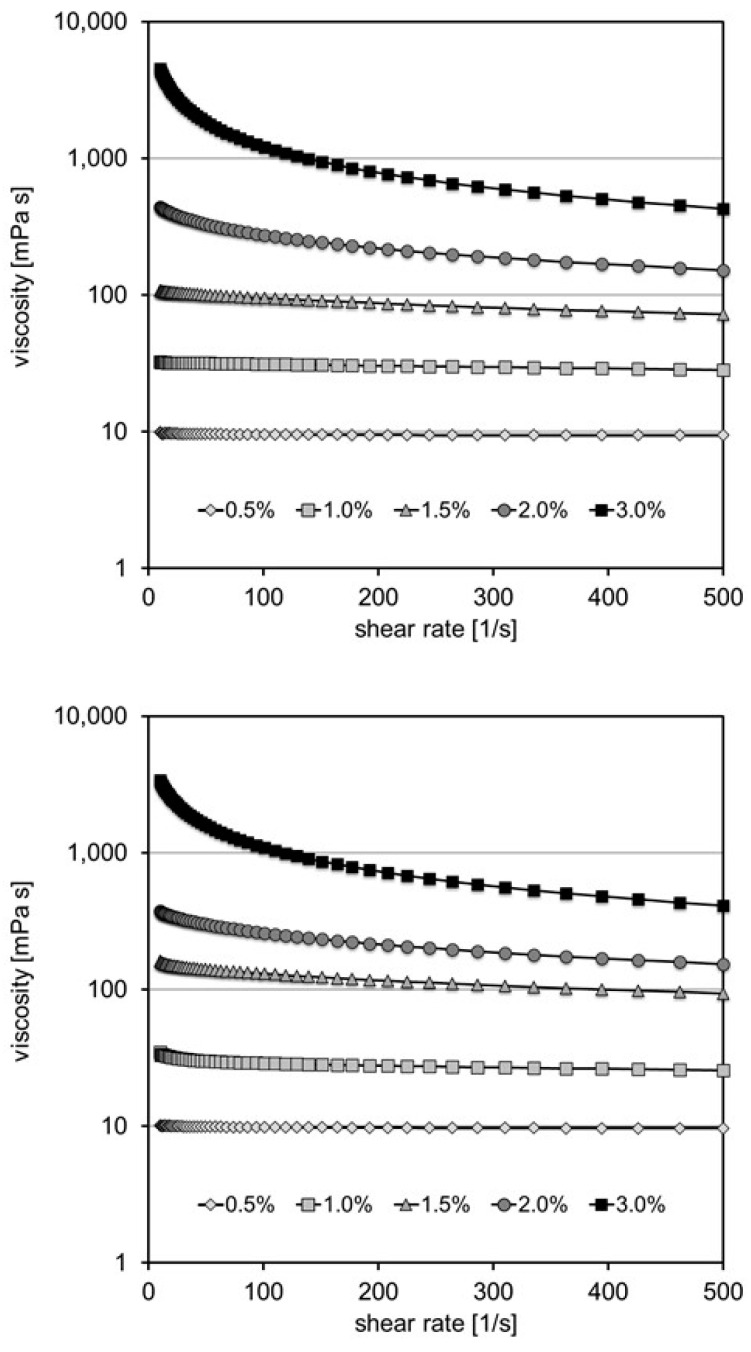
Viscosity curve for pectin with a degree of methyl-esterification of 30% (P30; top) and a degree of methyl-esterification of 35% (P35; bottom) in aqueous solution at a content of 0.5–3.0 wt.%.

**Figure 2 foods-13-02885-f002:**
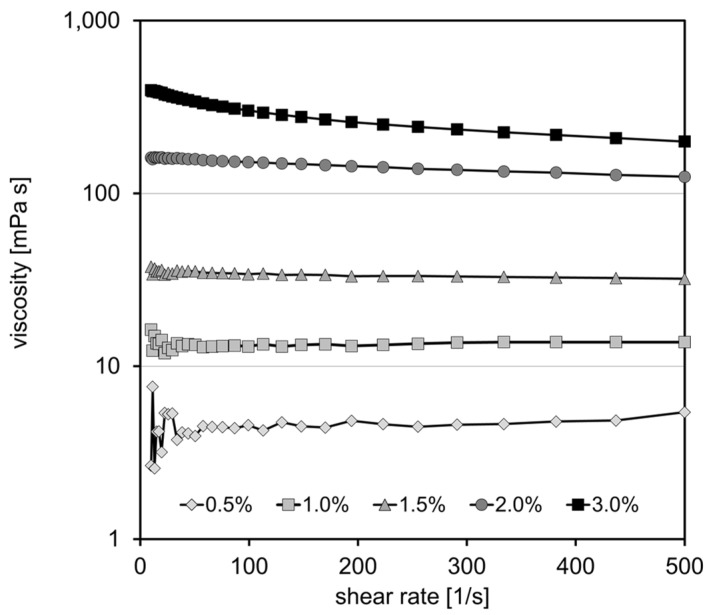
Viscosity curve of chitosan CsH in aqueous solution with 0.5 wt.% glacial acetic acid at a content of 0.5–3.0 wt.%.

**Figure 3 foods-13-02885-f003:**
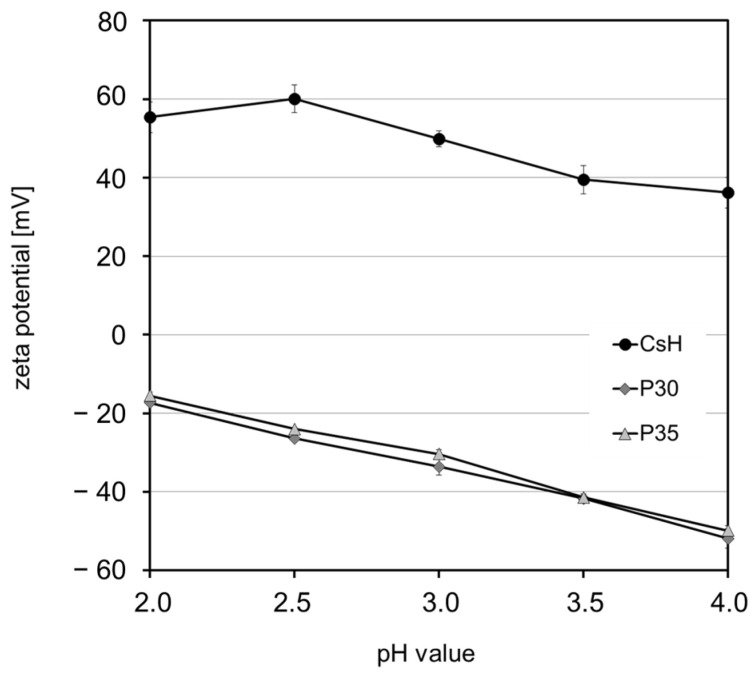
Zeta potential of pectin with a degree of methyl esterification of 30% (P30) or 35% (P35) and chitosan (CsH) in aqueous solution.

**Figure 4 foods-13-02885-f004:**
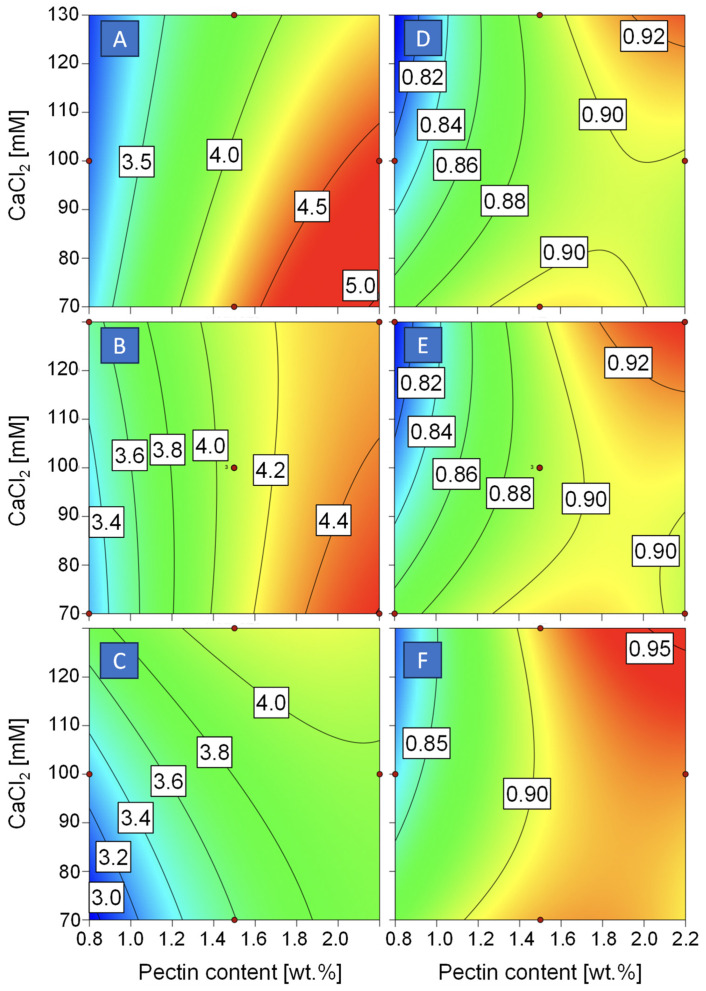
Impact of pectin content and calcium chloride concentration (CaCl_2_) on mean hydrogel bead size (**A**–**C**) and sphericity (**D**–**F**) of pectin–chitosan beads at a chitosan content of 0.8 wt.% (**A**,**D**), 1.5% wt.% (**B**,**E**), or 2.2 wt.% (**C**,**F**).

**Figure 5 foods-13-02885-f005:**
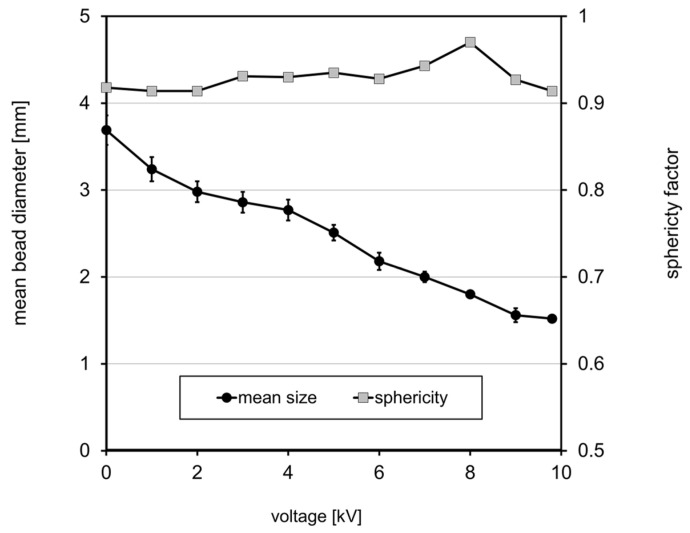
Impact of applied voltage during electrostatic droplet generation on bead size and sphericity for pectin–chitosan beads (2.2% pectin and 2.2% chitosan, extruded into a 130 mM calcium chloride solution).

**Figure 6 foods-13-02885-f006:**
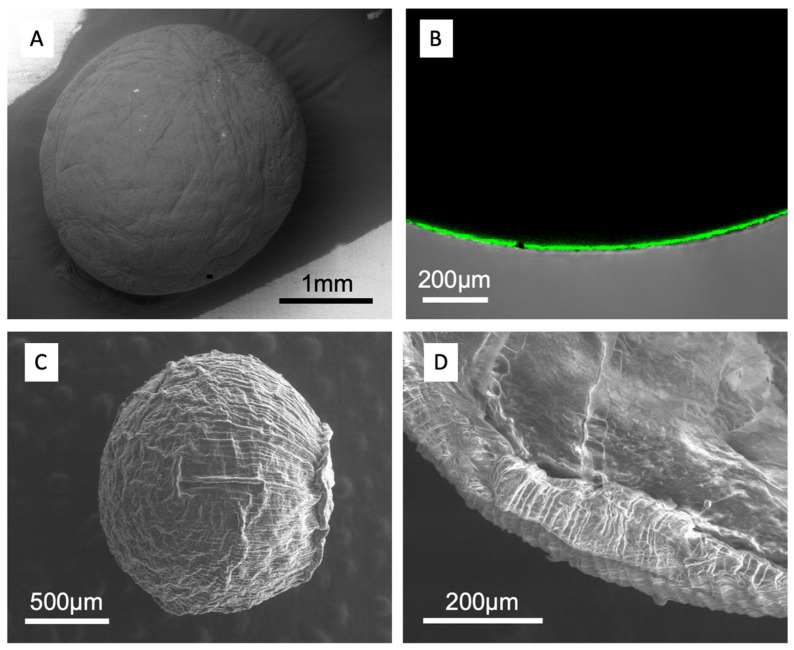
Microbead structure as revealed by scanning electron microscopy at varied magnifications (**A**,**C**,**D**) and and confocal laser scanning microscopy (**B**) where green colour indicates the presence of chitosan. (2.2 wt.% pectin solution extruded into an aqueous calcium chloride solution (130 mM) containing 2.2 wt.% chitosan CsH).

**Figure 7 foods-13-02885-f007:**
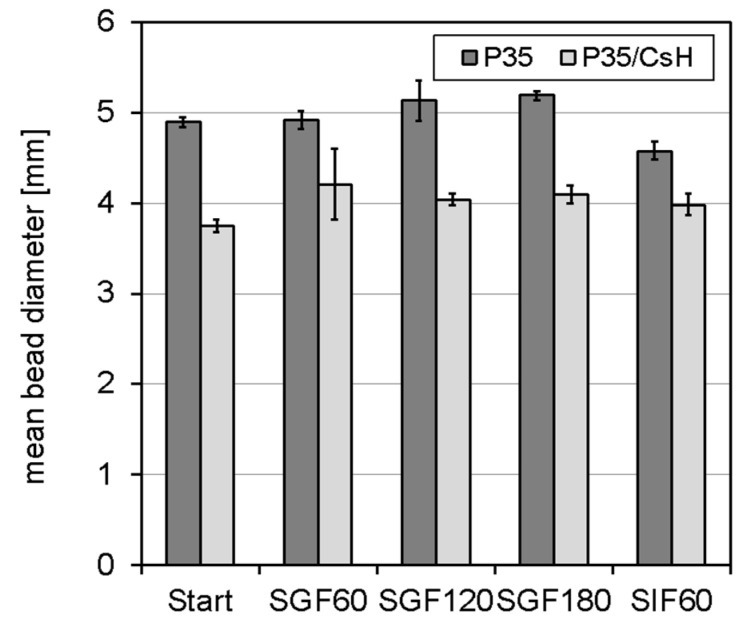
Mean particle diameter of hydrogel beads (2.2 wt.% pectin solution extruded into an aqueous calcium chloride solution (130 mM) with or without 2.2 wt.% chitosan CsH) during incubation in simulated gastrointestinal fluids (SGF: simulated gastric fluid, SIF: simulated intestinal fluid, numbers indicate the duration of incubation in min).

**Table 1 foods-13-02885-t001:** Independent variables and working levels used in the Box-Behnken design to optimize conditions for the elaboration of pectin-chitosan beads and to examine their effects.

Independent Variables	Level and Value		
	−1	0	+1
Pectin (wt.%)	0.8	1.5	2.2
Chitosan (wt.%)	0.8	1.5	2.2
Calcium (mM)	70	100	130

**Table 2 foods-13-02885-t002:** ANOVA for the linear model of a Box–Behnken design investigating the impact of chitosan and pectin content and calcium chloride concentration on mean particle size.

Source	Sum of Squares	dF	Mean Square	F-Value	*p*-Value
Model	2.41	3	0.8035	9.72	0.0020
A-Pectin	2.29	1	2.29	27.74	0.0003
B-Chitosan	0.1162	1	0.1162	1.40	0.2609
C-CaCl_2_	0.0002	1	0.0002	0.0029	0.9578
Residual	0.9095	11	0.0827		
Lack of fit	0.8293	9	0.0921	2.30	0.3402
Pure error	0.0803	2	0.0401		
Cor Total	3.32	14			

**Table 3 foods-13-02885-t003:** ANOVA for the quadratic model of a Box–Behnken design investigating the impact of chitosan and pectin content and calcium chloride concentration on particle sphericity.

Source	Sum of Squares	dF	Mean Square	F-Value	*p*-Value
Model	0.0192	9	0.0021	15.74	0.0037
A-Pectin	0.0126	1	0.0126	93.06	0.0002
B-Chitosan	0.0005	1	0.0005	3.54	0.1188
C-CaCl_2_	0.0002	1	0.0002	1.47	0.2792
AB	0.0000	1	0.0000	0.3110	0.6011
AC	0.0029	1	0.0029	21.07	0.0059
BC	0.0000	1	0.0000	0.3110	0.6011
A2	0.0022	1	0.0022	16.37	0.0099
B2	0.0001	1	0.0001	0.4259	0.5428
C2	0.0005	1	0.0005	3.57	0.1175
Residual	0.0007	5	0.0001		
Lack of fit	0.0005	3	0.0002	2.30	0.3175
Pure error	0.0002	2	0.0001		
Cor Total	0.0199	14			

## Data Availability

The original contributions presented in the study are included in the article, further inquiries can be directed to the corresponding author.

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
