# Peer review of "Pectin–Chitosan Hydrogel Beads for Delivery of Functional Food Ingredients"

_foods, 2024, doi:10.3390/foods13182885_

Round 1
Reviewer 1 Report
Comments and Suggestions for Authors
This study investigated the impact of formulation and process conditions on the size, sphericity, and dissolution behavior of chitosan-coated hydrogel beads prepared by interfacial coacervation. The size and sphericity of the beads depend on the formulation and range from approximately 3-5 mm and 0.82-0.95, respectively. Process conditions during electro-dripping may be modulated to tailor bead size. Depending on the voltage, bead size ranged from 1.5-4 mm. The following issues should be addressed.
1. Authors stated that the CLSM and SEM confirmed chitosan shell formation around the pectin bead. However, the shell thickness looks not the same from CLSM and SEM.
2. Where is the figure 3?
3. The delivery and release of functional food ingredients were not studied in the study, which should be added.
Reviewer 2 Report
Comments and Suggestions for Authors
The manuscript entitled ‘Pectin-Chitosan Hydrogel Beads for Delivery of Functional Food Ingredients’ presents an investigation into the development of pectin-chitosan hydrogel beads for the delivery of functional food ingredients. The topic is timely and relevant, given the growing interest in functional foods and the need for innovative delivery systems to enhance the stability and bioavailability of bioactive compounds. The manuscript is well-organized, with a clear structure and logical flow of ideas. However, there are several areas that require improvement to strengthen the overall quality and impact of the paper.
1. The prepared pectin-chitosan beads should be characterized through FTIR to confirm the chemical interaction between the two components in the formation of beads.
2. The abstract should provide specific quantitative results, such as mention the encapsulation efficiency, release profiles, and others, to give readers a more comprehensive overview of the study's outcomes.
3. The introduction outlines the importance of functional food ingredients and the potential of pectin-chitosan hydrogels as delivery systems. However, the literature review could be expanded to include more recent studies that have explored similar systems.
4. The research gap and the objectives of the study are well-stated, but the novelty of the study is not clearly highlighted. It is recommended that the author emphasize what sets this study apart from previous research in the same area.
5. Did you determine the surface porosity of the beads? Were the beads porous or smooth? This investigation will be important for the drug loading efficiency.
6. The methodology section should provide clear information on the preparation method of the hydrogel beads like the specific concentrations of pectin and chitosan used, as well as the conditions, for example, pH and temperature used for preparation of beads.
7. The discussion could be more in-depth by providing a more critical analysis of the results, comparing them with existing literature and discussing any discrepancies or unexpected findings.
8. The release profiles of the functional ingredients from the hydrogel beads are discussed, but the mechanisms underlying the release behavior should be explored in more detail. What factors (e.g., diffusion, degradation) are driving the release, and how do these compare to other similar systems?
9. What the differences in encapsulation efficiency or release rates imply for the practical application of these hydrogel beads in the food industry?
10. The conclusions section should provide specific implications for future research or potential commercial applications.
11. The manuscript is generally well-written but could benefit from a thorough proofreading to correct minor grammatical errors and improve sentence structure. For instance, some sentences are long and complex, making them difficult to follow.
12. Additional comments/corrections/suggestions:
- Perhaps Highlights are not required by the MDPI journals and should be removed.
- Define all abbreviations at first appearance and then used consistently.
- Merge figure 1 and 2 as both present data on the viscosity curve for pectin and chitosan, respectively. This will provide a better comparison before combining them.
- Table caption: Correct Tab. to ‘Table’ (line 764 and 769). Change colon (:) to dot (.) in table and figure captions.
- Insert space between units and values except for % and temperature.
- Change %w/w to ‘wt.%’.
Comments on the Quality of English LanguageMinor language editing
Reviewer 3 Report
Comments and Suggestions for Authors
The manuscript describes pectin-chitosan microparticles for oral delivery of functional food ingredients. The authors attempted to describe the phase behavior and rheology of oppositely charged polyelectrolytes toward the formation of stable hydrogel beads. The beads produced were studied under gastrointestinal conditions for swelling and maintaining structural integrity. The manuscript has the potential to contribute to the expansion of knowledge on biopolymer-based hydrogel microcarriers and seems to be interesting, but the topic has been treated superficially so far. Therefore, I would like to draw attention to some issues that the authors should address in order to increase the scientific contribution of this manuscript.
- There is a lack of a clear statement of novelty. The article in this form appears to be a generic article on the pectin-chitosan delivery system.
- As for a research paper, there is too much information in the Introduction and Conclusion sections recalled from different articles. The authors should focus more on how the physicochemical properties of the polyelectrolytes affect the hydrogel under specific conditions, rather than reviewing the already described alginate-chitosan delivery systems.
- Please explain what is meant by: " P30 showed a calcium sensitivity of 253.8 mPa s, while P35 showed a calcium 119 sensitivity of 476.5 mPa s (...)" (lines 119–120). This parameter describes the viscosity of the solutions and not the ability of the polymers to complex calcium ions.
- Please provide further details about the pectins used in the experiments, such as their origin, degree of esterification, monosaccharide composition, molecular weight, etc.
- There should be an illustrative diagram of the “electro-droplet system” used to form pectin droplets, since the authors claim that this equipment was self-made.
- Lines 282–283: The authors should use a higher concentration of P30 than 1.5% to confirm that P30 cannot form calcium-pectate spheres.
- Provide an explanation for why the thickness of the chitosan shell is different when observed by confocal microscopy and SEM (Figure 6).
- Is there any particular reason why the core separates from the shell visible in Figure 6D?
- To describe the structural changes of the hydrogel spheres and the integrity of the chitosan shell associated with swelling under SGF and SIF conditions, visual examination should be performed by confocal microscopy and/or SEM at predetermined time points, i.e. 60, 120, 180 min.
- The P35/CsH system cannot be called the 'intestinal delivery system' (line 465) because it disintegrated in the simulated gastric fluid. The typical residence time of food in the upper GI tract is 120 minutes, so the studied microparticles will not be able to transport their payload to the intestines.
Comments on the Quality of English LanguageThe manuscript requires thorough language editing, because in this form it is difficult to follow the authors' reasoning.
Reviewer 4 Report
Comments and Suggestions for Authors
The aim of this paper was to prepare chitosan-coated pectin hydrogel beads for intestinal delivery of functional food ingredients. Also, the impact of formulation and process conditions on the size, sphericity, and dissolution behavior of chitosan-coated hydrogel beads was examined.
Manuscript fits the journal scope. It is overall well written and comprehensive. The used literature is adequate and the applied methods are well described. The obtained results are relevant and well presented. The discussion is presented simultaneously with the results.
Suggestions to the authors:
Comment 1
It is necessary to explain the meaning of every abbreviation used in the abstract (e.g. CLSM, SEM) as well as in the entire manuscript (e.g. SGF, SIF).
Comment 2
Page 4, Table 1.
The value of the independent variable chitosan (%) at the level -1 should be 0.8. Please, revise it.
Comment 3
Page 5, lines 229-230.
Hydrogel bead preparation was not described in paragraph 2.5. Please, revise.
Comment 4.
Page 6 and 7, Figure 1.
Paragraph 2.2. Viscosity Measurements (lines 136-142) stated that used shear rate was is the range of 0-500 s-1, while x-axis in Figure 1 has range of 0-30 s-1. Please, revise it.
Comment 5.
Page 8, lines 275-277.
The Figure name and numeration are not correct. Please, revise.
Comment 6.
Page 13, lines 394-396.
Please revise the following sentences (since Figure 7 does not show the stated ranges and the change exists although probably it is not significant):
“Fig. 7 shows that particle size was in a similar range of 4 to 4.5 mm, and the sphericity factor ranged from 0.93 to 0.95. Furthermore, mean particle size did not change in both samples during incubation in simulated gastric fluid (pH 3) for 180 min.”
Comment 7.
Page 15, paragraph 4. Conclusions.
Please consider rewriting this paragraph in order to avoid analysis of the other authors’ results in the conclusion.
Round 2
Reviewer 1 Report
Comments and Suggestions for Authors
NO COMMENTS
Reviewer 2 Report
Comments and Suggestions for Authors
Accept
Comments on the Quality of English LanguageMinor language editing needed